Arbuscular mycorrhizal fungi enhance drought resistance in Bombax ceiba by regulating SOD family genes

Luo Changxin 1
Li Zhumei 1
Shi Yumei 1
Gao Yong 1
Xu Yanguo 1
Zhang Yanan 1 yananzh@mail.qjnu.edu.cn
Chu Honglong 1 2 honglongchu@mail.qjnu.edu.cn
1 College of Biological Resource and Food Engineering, Center for Yunnan Plateau Biological Resources Protection and Utilization, Qujing Normal University , Qujing, Yunnan , China
2 Qujing Normal University, Key Laboratory of Yunnan Provincial Department of Education of the Deep-Time Evolution on Biodiversity from the Origin of the Pearl River , Qujing, Yunnan , China
Root-Bernstein Meredith
Electronic publication date: 2024 Aug 7
Publication date: 2024
Volume: 12
Electronic Location ID: e17849
Received 2024 Feb 23; Accepted 2024 Jul 10
Copyright: © 2024 Luo et al.
Copyright year: 2024
Copyright holder: Luo et al.
License: This is an open access article distributed under the terms of the Creative Commons Attribution License, which permits unrestricted use, distribution, reproduction and adaptation in any medium and for any purpose provided that it is properly attributed. For attribution, the original author(s), title, publication source (PeerJ) and either DOI or URL of the article must be cited.
License URL: https://creativecommons.org/licenses/by/4.0/

Keywords: Drought resistance, AMF, Superoxide dismutase (SOD) genes, Genomic analysis, Reactive oxygen species (ROS) scavenging

Funding: National Natural Science Foundation of China 31860057 Yunnan Applied Basic Research Projects 202401AT070002 This work was supported by the National Natural Science Foundation of China (No. 31860057) and the Yunnan Applied Basic Research Projects (No. 202401AT070002). The funders had no role in study design, data collection and analysis, decision to publish, or preparation of the manuscript.

==============================
The physiological activity facilitated by arbuscular mycorrhizal fungi (AMF) contributes to plants’ ability to tolerate drought. Nevertheless, it is unclear if AMF colonization affects the expression of genes in the host plant that encode antioxidant enzymes in the superoxide dismutase (SOD) family, which help alleviate drought stress in plants. Here, we conducted a pot trial to determine whether colonization by the AMF Rhizophagus irregularis improves drought resistance in Bombax ceiba. We comprehensively analyzed the SOD gene family and evaluated genome-wide expression patterns of SODs and SOD activity in AMF-colonized and non-mycorrhizal plants under simulated drought. We identified a total of 13 SODs in the genome of B. ceiba, including three FeSODs (BcFSDs), three MnSODs (BcMSDs), and seven Cu/ZnSODs (BcCSDs). Phylogenetic analysis based on binding domain revealed that SOD genes from B. ceiba and various other plant species can be divided into three separate groups, showing significant bootstrap values. Our examination of gene composition and patterns suggests that most BcSOD genes in these three subgroups are significantly conserved. Additionally, it was noted that hormones and stress-responsive cis-regulatory elements were found in all BcSOD promoters. Expression profiling by qRT-PCR demonstrated that AMF increased relative expression levels of Cu/Zn-SODs in both roots and shoots under drought stress, except for BcCSD3 in roots. Furthermore, AMF colonization increased the relative expression of BcMSD1a and BcMSD1b in roots, augmenting SOD activities and increasing ROS scavenging during drought. In general, this work offers molecular evidence in support of the beneficial effect of AMF colonization on drought tolerance in B. ceiba. It also elucidates the expression patterns of SOD genes, which will support efforts to optimize mycorrhizal seedling cultivation under stressful conditions.

Introduction

As climate change increases the frequency and severity of drought worldwide, the influence of human activity continues to expand, contributing to ecological degradation both directly and indirectly (Trenberth et al., 2014). Drought stress can trigger the accumulation of reactive oxygen species (ROS) within plant cells (Tiwari et al., 2017; Hasanuzzaman et al., 2020), causing oxidative damage and reducing plant biomass and ecosystem productivity (Raja et al., 2017; Choudhary et al., 2018). The harmful effects of drought on plants can thus be partly attributed to the buildup of ROS (Zou, Wu & Kuča, 2021; Yang et al., 2023). Controlling scavenging is essential to avoid ROS accumulation and to enhance stress resistance. Plants have developed intricate and extensive mechanisms to defend themselves from the harmful effects of ROS accumulation using a range of both enzymatic and non-enzymatic antioxidant compounds (Mittler, 2017; Batool et al., 2019; Hasanuzzaman et al., 2020; Gui et al., 2024). Superoxide dismutase (SOD) is an antioxidant enzyme that acts as the main scavenger of ROS and has the crucial role of controlling many of the physiological and biological mechanisms that plants use to cope with environmental stress (Mittler, 2017; Ullah et al., 2024). SODs are a class of metal-binding enzymes that convert the superoxide anion (O2−), a central component of ROS, into hydrogen peroxide (H2O2) and oxygen. This transformation is accomplished via disproportionation and is catalyzed by a metal cofactor (Sies, Berndt & Jones, 2017). Under biotic and abiotic stress, SOD can remove excess ROS during the rapid growth of plant organs, safeguarding cells from oxidative damage and stabilizing physiological metabolism (Hodgson & Fridovich, 1973; Mittler, 2017).

SODs are encoded by a diverse family of genes that contain sequences for many other metalloenzymes, which are biochemically similar but molecularly diverse. These genes can be classified into three groups, defined by the identity of the enzyme’s metal cofactor: copper/zinc SOD (Cu/ZnSOD), manganese SOD (MnSODs), and iron SOD (FeSOD) (Zelko, Mariani & Folz, 2002; Mittler, 2017). Nickel SOD (NiSOD) is mainly present in marine organisms and bacteria, including cyanobacteria and Streptomyces, but has not been detected in plants (Schmidt et al., 2009). SOD proteins are commonly found across a range of cellular organelles within plants, and their subcellular localization is typically linked to places where O2− is produced. The effective positioning of SOD proteins in cellular compartments facilitates the conversion of cellular signals and is thus crucial for efficient responses to abiotic oxidative stress (Del Río et al., 2018). Fe-SODs can be found in chloroplasts, while Mn-SODs are present in mitochondria and peroxisomes (Abreu & Cabelli, 2010; Perry et al., 2010). Cu/Zn-SODs are widely distributed across organelles but are mainly found in chloroplasts, cytoplasm, and peroxisomes (Song et al., 2018).

Over the past few years, numerous studies have demonstrated that various types of environmental pressure can elicit reactions in the transcription levels of plant SOD genes. These reactions help plants cope with stress caused by extreme temperatures, high salinity, drought, and other extreme conditions (Mosa et al., 2018; Chokshi et al., 2020; Liu et al., 2021). Recent work indicates that increased SOD activity and higher expression of genes that encode SOD are associated with improved plant tolerance to different types of stress (Pour-Aboughadareh et al., 2020; Su et al., 2021; Yang et al., 2023).

Arbuscular mycorrhizal fungi (AMF) are members of the phylum Glomeromycotina that form mutually beneficial relationships known as arbuscular mycorrhiza (AM) with more than 80% of land-dwelling plants (Bonfante, 2018). This association is generally recognized as advantageous for both partners, with some work suggesting that it was crucial early in terrestrial colonization by plants (Wipf et al., 2019; Genre et al., 2020). AMF play vital roles in a diversity of functional processes, including carbon metabolism, nutrient acquisition, water absorption, and resistance to abiotic and biotic stress (Genre et al., 2020; Shi et al., 2021; Li et al., 2022). Colonization by AMF enhances the growth and stress tolerance of host plants (Ruiz-Lozano et al., 2016; Zhang et al., 2019; Li et al., 2022). AM symbiosis develops as arbuscules form inside the cells of host plants, acting as the locus for symbiont-host communication and nutrient exchange (Etemadi et al., 2014). Some work reports elevated expression of catalase and peroxidase genes in addition to high concentrations of hydrogen peroxide in plant cells with arbuscules (Lambais, Ríos-Ruiz & Andrade, 2003). However, how the host plant’s AMF partner responds to drought stress, especially in relation to the SOD gene family, remains poorly understood.

Bombax ceiba Linn. is a versatile tree species with economic and ecological significance (Gao et al., 2018). Many countries throughout Asia rely on this species to provide nourishment, medication, textiles, energy, animal feed, and numerous other essential commodities (Jain & Verma, 2012; Gao et al., 2018). Futhermore, B. ceiba is a pioneer species widely planted as part of reforestation efforts in arid valleys thanks to its ability to thrive in areas with little rainfall and good drainage (Jin, Yang & Tao, 1995). The hardiness of this species makes it well-suited for use in ecological restoration efforts, and it is a potential candidate for reforestation projects in dry and semi-dry regions in tropical and subtropical zones. Consequently, expanding this species’ cultivation represents an opportunity to simultaneously promote conservation and economic growth.

In this study, a thorough analysis of the entire genome was conducted in order to discover superoxide dismutase genes (SODs) in B. ceiba. Furthermore, the phylogenetic relationships, preserved patterns, genetic arrangements, and regulatory elements were characterized. Additionally, we evaluated expression profiles of the whole SOD gene family and SOD activities in the roots and shoots of plants with and without AM associations under drought stress. This work improves understanding of SOD genes in B. ceiba, providing valuable insights into how plant SOD genes may enhance drought tolerance and resistance through AMF-mediated responses.

Materials and Methods

Experimental design

We employed a fully randomized block design with two variables: (1) AMF treatment: presence (+AMF) or absence (−AMF) of Rhizophagus irregularis; and (2) soil water regime: well-watered (WW) at 60% water holding capacity or drought stress (DS) at 30% water holding capacity. Each replicate unit consisted of three pots, and each treatment was replicated four times.

Mycorrhizal inoculum

The AMF treatment group was inoculated with R. irregularis fungus (Glomeromycetes; Glomerales; Glomeraceae; Rhizophagus), which was obtained from the Beijing Academy of Agriculture and Forestry Sciences. It was propagated with corn (Zea mays L.) and Trifolium repens L. at the Center for Yunnan Plateau Biological Resources Protection and Utilization, Qujing Normal Uni-versity (Qujing, China). Propagule density was estimated at 226 propagules/mL using the most likely number method (Feldmann & Idczak, 1992). During transplanting, seedling roots were treated with 10 mL mycorrhizal inoculum. Control seedings received 10 mL of sterilized inoculum and 10 mL of inoculum washing solution obtained from live inoculum, which had previously been filtered through a 1 μm nylon mesh.

Plant culture

After collection from a hot-dry valley (25°40′50.06″N and 101°53′27.76″E), the surfaces of B. ceiba seeds were sterilized by soaking in a 30% hydrogen peroxide solution for 30 min and then rinsed five times with sterile water. To initiate pre-germination, sterilized seeds were incubated on sterile gauze in Petri dishes (15 cm) at ambient temperature. Seeds were washed twice daily with sterilized water. Following germination, seeds were transferred to incubation plates containing autoclaved vermiculite and incubated at 25 °C, with a photoperiod of 14 h of light and 10 h of darkness. Similarly sized seedlings (~6–7 cm) were selected and transplanted into plastic pots (26 cm diameter, 19.5 cm height). Each pot contained 8 kg of sand, which had been washed five times with tap water and autoclaved at 121 °C for 2 h for use as a substrate.

Soil water regimes

Soil moisture in WW and DS treatments was maintained at 60% and 30% of substrate water holding capacity, respectively, after Li et al. (2022). Prior to drought simulation, well-watered conditions were maintained for all pots. We began the drought simulation 4 weeks after seedlings were transplanted. Pots were weighed daily and watered as needed to maintain stable soil moisture levels for 45 days. Plants were grown in a controlled environment with temperatures ranging between 24 °C and 32 °C. They were exposed to 14 h of daylight, and humidity was maintained between 40% and 60%. Every 10 days, all pots received 100 mL of Hoagland solution for irrigation (Hoagland & Arnon, 1950).

Plant sampling and biomass measurement

At the end of the 45-day drought simulation, seedling height and basal diameter were measured using a ruler and a vernier caliper, respectively, and stems and roots were collected from each plant. To evaluate AMF colonization, roots were washed with tap water, and a little of them were conserved in FAA fixative (37% formaldehyde, glacial acetic acid, and 95% ethanol, 9:0.5:0.5 (v:v:v)). The remaining portion was flash frozen using liquid nitrogen and stored at −80 °C for future analysis.

Root staining and evaluation of mycorrhizal colonization

Root samples were stained using trypan blue, after Phillips & Hayman (1970). Prior to staining, roots were acid treated. They were first submerged in 10% KOH solution at 90 °C for 30 min before treatment with 10% H2O2 at ambient temperature for 10 min, followed by immersion in 2% hydrochloric acid for 5 min. Roots were then stained using trypan blue for 30 min at 90 °C. To evaluate rates of mycorrhizal colonization of hyphal, arbuscular, vesicle, and spore structures, the magnified cross-sections were utilized (McGonigle et al., 1990) using a Leica DM2500 microscope (CMS, GmbH, Wetzlar, Germany).

ROS and lipid peroxidation measurement

Powdered samples were mixed with extraction buffer (1% PVP, 1 mM EDTA, and 50 mM potassium phosphate buffer to achieve uniformity) and centrifuged at 14,000 g for 30 min at 4 °C. Soluble protein and SOD activity in the resulting supernatant was quantified using the nitroblue tetrazolium reduction approach with photochemical inhibition using the procedure described by Beyer & Fridovich (1987).

Identification of SOD family genes across the B. ceiba genome

The entire genome of B ceiba and associated protein sequences were retrieved from the GigaDB Dataset (http://gigadb.org/dataset/100445) (Gao et al., 2018). To identify SOD-encoding genes, nine reported SOD protein sequences from Arabidopsis thaliana were employed as query sequences in a local BLASTP search against the B. ceiba genome, with an E-value cutoff of 1.0 × e−5. Candidate SODs were selected based on E-values (<1.0 × e−10) and sequence homology values (>60%). Corresponding cDNA and protein sequences for the selected candidates were extracted. All candidate sequences that meeting the criteria were further validated using the InterPro (https://www.ebi.ac.uk/interpro/) database, the Hammer SMART (http://smart.embl-heidelberg.de/#) database, and NCBI Conserved Domain Search (https://www.ncbi.nlm.nih.gov/Structure/cdd/wrpsb.cgi) for conserved domain verification. Databases were queried for SOD domains, such as SOD_Fe_C-terminal (PF02777), iron/manganese SOD, SOD_Fe_N-alpha-hairpin (PF00081), and Copper/zinc SOD, Cu/Zn SOD (PF00080.23). Sequences containing the relevant domains were classified as BcSODs.

Redundant sequences were eliminated via alignment using ClustalW. Subsequent analyses employed unique sequences. Phylogenetic analysis was conducted for candidate SOD genes in B. ceiba and compared with SOD genes in Arabidopsis thaliana and various other plant species. We employed the naming conventions used for A. thaliana SOD genes (Table 1).

Table 1 Colonization rate of B. ceiba seedlings.

Treatment	Colonization rate (%)	
Hypha	Arbuscule	Vesicle	Spore	
Well-watered	74.40 ± 1.64	50.49 ± 2.68	5.65 ± 0.41	48.33 ± 4.23	
Drought stress	53.79 ± 5.86	41.19 ± 6.59	25.06 ± 3.31	37.74 ± 1.49	
P-value	**	*	***	**	
F-value	45.894	7.917	135.478	22.298	
Note:

The data are the means of four biological replicates ±SD (n = 4). Significant difference between well watered and drought stress was tested by Student’s T test. *: 0.01 < P < 0.05, **: 0.001 < P < 0.01, ***: P < 0.001. Hypha, arbuscule, vesicle and spore are structure of AMF.

SOD protein properties, subcellular location, multiple sequence alignment, and phylogenetic analysis

Physicochemical properties, such as amino acid molecular weights, lengths, and isoelectric points of SOD family members were predicted using the ProtParam function in the Expasy online platform (http://web.expasy.org/protparam/). Subcellular localization of SOD protein family members was predicted using WoLFPSORT (https://wolfpsort.hgc.jp/).

Multiple amino acid sequences of BcSODs were aligned and colored using DNAMAN7.0 (http://www.lynnon.com/) (see Fig. S1). To analyze phylogenetic relationships between SOD proteins in different plants, full-length amino acid sequences of 35 SOD proteins from Arabidopsis, Oryza sativa, Populus trichocarpa, and Dimocarpus longan were used to construct a phylogenetic tree. Trees were constructed using ClustalW in MEGA 7 (http://www.megasoftware.net/) with default parameters, employing the maximum likelihood method. We combined the JTT matrix-based model with pairwise gap deletion. A total of 1,000 bootstrap replicates were carried out.

Protein-protein interaction analysis of BcSODs

The apparent physical protein-protein interactions of BcCSD, BcMSD and BcFSD were estimated using STRING v9.0 (http://string-db.org/), and the associated graphic was generated using cytoscape (version 3.10.1, https://Cytoscape.org/) (Otasek et al., 2019).

Conserved motif, gene structure, and promoter analysis of BcSODs

The discovery of conserved motifs among the 13 BcSOD proteins was performed using the Multiple Em for Motif Elicitation (MEME) (http://meme-suite.org/tools/meme) tool, with motif number set to 12 and motif width ranging from 6 to 50. Structure conservativeness was predicted for BcSODs using the NCBI website. Downloaded files were visualized using the “Visualize NCBI CDD Domain Pattern” function in TBtools 2.019 (Chen et al., 2020). To predict potential stress-responsive cis-elements in the promoter regions of BcSOD genes, 3,000 bp long sequences were extracted upstream from the transcription start sites for each gene using GFF3/GTF Manipulate in the Sequence Toolkit and Fasta Extract in TBtools 2.019. These sequences were considered to represent the promoter sequences. The distribution and types of cis-acting elements in the promoters were subsequently analyzed using the PlantCARE website. Results were visualized using the Biosequence Structure Illustrator function of TBtools 2.019.

Gene expression profile of BcSODs in B. ceiba

Total RNA was extracted from shoots and roots using a Plant RNA extraction kit (Omega Bio-Tek, Norcross, GA, USA) according to the manufacturer’s instructions. RNA concentrations were quantified using a NanoDrop 2000 (Thermo Scientific, Waltham, MA, USA), and RNA quality was confirmed using agarose gel electrophoresis. First-strand cDNA synthesis was performed using a PrimerScript® RT Reagent Kit with gDNA Eraser (TaKaRa Bio, Dalian, China) according to the supplier’s protocol.

Root and shoot BcSOD expression in response to drought stress was profiled using quantitative real-time PCR (qRT-PCR) using a Roche LightCycle 96 machine (Roche, Basel, Germany) with SYBR Green qPCR kits (TaKaRa, Dalian, China) according to the manufacturer’s instructions. The Actin gene was used as the internal standard (Li et al., 2022). Gene-specific primers were designed for these amplifications (Table S1). Each reaction included 10.0 µL of SYBR R Premix Ex Taq TM (TaKaRa, Dalian, China), 4.0 µL of tenfold diluted cDNA as a template, 1 µL of each specific primer, and 4 µL of ddH2O, for a total volume of 20 µL. Thermal cycles were as follows: an initial step at 95 °C for 3 min, followed by 40 cycles of 95 °C for 20 s, 56 °C for 20 s, and 72 °C for 20 s. Each qRT-PCR was performed three times using four separate RNA extracts from four biological replicates to minimize inherent errors. The relative expression levels of all BcSOD genes were calculated using the 2−ΔΔCT method (Livak & Schmittgen, 2001).

Results

Root colonization and growth conditions

Prominent mycorrhizal formations, including arbuscules, vesicles, and inter-radical spores, were observable in the roots of B. ceiba plants from both water regime treatments (Figs. 1A–1C and Table 1), indicating that they had been colonized by AMF (R. irregularis). As anticipated, plants that were not inoculated (NMF) showed no indication of mycorrhizal colonization in either water regime treatment (Fig. 1A). Drought stress caused a significant reduction in hypha and arbuscule colonization (38.58% and 22.01%, respectively), while simultaneously increasing spore and vesicle colonization (44.25% and 28.67%, respectively) compared to well-watered (Table 1).

Figure 1 Root colonization condition of AMF.

(A) Image of Bambax ceiba root non-inoculated with AMF; (B and C) Image of Bambax ceiba root inoculated with AMF. The blue and dark blue parts in the picture are the structures of AMF stained with trypan. The red arrows in (B) and (C) refer to spores, vesicle, hyphae, and arbuscule, respectively. All images used in this figure were taken by the authors during the experiment process and are original works.

Seedling growth was significantly affected by the combined influence of drought stress and AMF colonization. Drought stress reduced seedling height, but AMF colonization significantly increased height and basal diameter in both the DS (49.39% and 45.59%, respectively) and the WW (45.64% and 59.89%, respectively) treatment groups compared to NMF (Figs. 2A–2C). In general, AMF plants had better growth than NMF plants in both DS and WW (Fig. 2A). Furthermore, SOD activity was higher in AMF roots and shoots in both water regime treatments. Under drought conditions, AMF colonization increased SOD notably in both roots and shoots. However, the influence of AMF colonization was minimal in the WW treatment group. Moreover, SOD activities were higher under DS relative to WW (Fig. 3).

Figure 2 Growth condition of B. ceiba.

(A) B. ceiba seedlings grow in pots received different treatments; (B) Height of B. ceiba seedlings; (C) Basal diameter of B. ceiba seedlings. The data presented are the means of four biological replicates ± standard deviation (n = 4). Significant differences among the means using Turkey’s test (P < 0.05) are indicated by distinct lowercase letters above each column. The two-way ANOVA had a notable impact with **: 0.001 < P < 0.01, ***: P < 0.001 denoting a highly significant effect, while NS indicated no significant effect. NMF indicate non-mycorrhizal treatment; AMF indicate arbuscular mycorrhizal fungi inoculation. All images used in this figure were taken by the authors during the experiment process and are original works.

Figure 3 The SOD levels in the roots and shoots of B. ceiba plants under conditions of drought stress.

Either the plants were treated with the AMF R. irregularis or they were not treated. The data presented are the means of four biological replicates ±SD (n = 4). Significant differences among the means, as determined by Turkey’s test (p < 0.05), are indicated by the distinct variations in lowercase and uppercase letters in the columns. The two-way ANOVA results showed a significant effect *: 0.01 < P < 0.05, ***: P < 0.001. NMF refers to the non-mycorrhizal treatment, while AMF represents the inoculation of arbuscular mycorrhizal fungi. DS indicates the drought-stressed treatment, with RDS referring to the root of the drought-stressed treatment and RWW representing the root of the well-watered treatment. LDS represents the shoot of the drought-stressed treatment, and LWW represents the shoot of the well-watered treatment.

Identification and physicochemical characterization of SOD proteins

Thirteen non-redundant putative BcSOD genes were successfully identified from the B. ceiba genome using local blastp searches. Potential matches, acquired via protein sequence analysis using documented SODs, were assessed for SOD domains using bioinformatics tools, including CDD, InterProScan, and Pfam. Our analysis identified a total of 13 SOD genes in B. ceiba (Table 2).

Table 2 Basic characteristics of the BcSOD gene and the encoded proteins.

Gene name	Sequence ID	Amino acid number	Molecular weight (kDa)	Theoretical PI	Instability index	Aliphatic index	GRAVY	Subcellular prediction	
BcCSD1a	Scaffold417.34	151	15.09	5.59	21.12	78.08	−0.098	cyto: 13	
BcCSD1b	Scaffold167.187	152	15.17	5.59	16.21	76.97	−0.144	cyto: 14	
BcCSD1c	Scaffold52.228	152	15.24	5.71	18.28	71.84	−0.216	cyto: 13	
BcCSD2a	Scaffold145.260	266	28.03	9.39	29.16	92.03	0.064	chlo: 7, extr: 3, E.R.: 2, mito: 1	
BcCSD2b	Scaffold25.390	214	22.47	9.12	24.96	86.12	−0.11	chlo: 9, mito: 5	
BcCSD3	Scaffold167.161	140	14.82	6.17	17.13	96.14	−0.128	cysk: 8, cyto: 4, extr: 1	
BcCSDc	Scaffold89.460	252	26.88	4.87	26.37	85.48	−0.194	pero: 11, cyto_nucl: 2, nucl: 1.5, cyto: 1.5	
BcFSD1	Scaffold63.64	306	34.91	4.92	44.57	78.46	−0.57	chlo: 14	
BcFSD2	Scaffold65.429	248	28.29	7.79	44.91	85.69	−0.321	chlo: 12, nucl: 1	
BcFSD3	Scaffold967.1	202	23.22	6.98	39.72	78.22	−0.484	chlo: 6, nucl: 3, cyto: 3, mito: 2	
BcMSD1a	Scaffold431.23	230	25.90	7.07	37.1	92.04	−0.37	mito: 9.5, cyto_mito: 5.5, chlo: 2, nucl: 1.5	
BcMSD1b	Scaffold101.63	235	26.27	7.1	32.64	93.45	−0.367	mito: 10, chlo: 3	
BcMSD2	Scaffold502.2	190	21.17	9.3	39.51	88.32	−0.467	mito: 8, chlo: 3, cyto: 1.5, cyto_nucl: 1.5, plas: 1	
Note:

cyto, cytoplasm; extr, extracellular; vacu, vacuole; E.R, Endoplasmic; mito, mitochondrio; cysk, cytoskeleton; golg, golgi apparatus; pero, peroxisome; plas, plasma membrane; chlo, chloroplasts; nucl, nucleus; GRAVY, grand average of hydropathicity.

Amino acid identity in the Zn/Cu-SODs subgroup ranged between 21.20% and 94.10% and between 23.00% and 95.40% in the Fe/Mn-SODs subgroup (Table S2). BcSOD proteins ranged in size from 140 to 306 amino acids, with calculated molecular weights between 14.82 and 34.91 kDa. Protein isoelectric points ranged from 4.87 to 9.39. Except for BcCSD2a, all BcSODs had negative Grand Average of Hydropathicity (GRAVY) values, ranging from −0.57 to 0.064. This suggests that they were primarily hydrophilic (Gasteiger et al., 2005). Instability index values suggested that all BcSODs except for BcFSD1 and BcFSD1 were unstable (Table 2). According to WoLFPSORT predictions, three members (BcCSD1a, BcCSD1b, and BcCSD1c) were located in the cytoplasm, while five members (BcFSD1, BcFSD2, BcFSD3, BcCSD2a, and BcCSD2b) were present in chloroplasts. All three MnSODs were found in mitochondria, BcCSDc was found in the peroxisome, and BcCSD3 was found in the cytoskeleton (Table 2).

InterPro and CDD were used to identify protein domains. Conserved domains, namely Mn/Fe_SOD_N and Mn/Fe_SOD_C, were detected in sequences containing BcMSD1a, BcMSD1b, BcMSD 2, BcFSD1, BcFSD2, and BcFSD3 (Figs. 4A–4C and S1A ). Furthermore, the SOD_Cu_Zn_dom domain was detected in various sequences, including BcCSD1a, BcCSD1b, BcCSD1c, BcCSD2a, BcCSD2b, BcCSD3, and BcCSDc (Figs. 4C and S1B). BcSODs were categorized into three separate clusters using phylogenetic connections and domain recognition: MnSOD, FeSOD, and Zn/CuSOD (Fig. 4A). This categorization demonstrates the structural and functional diversity present in the BcSOD gene family in B. ceiba.

Figure 4 Analysis of gene structures in BcSOD family.

(A) Phylogenetic tree of BcSOD genes. (B) Protein motifs in BcSODs. (C) Conserved domains found in BcSOD proteins. (D) Gene structures of BcSODs, including exons and introns. (A) The BcSODs were categorized into three separate groups based on the identification of domains and phylogenetic relationships by using One Step Build a ML Tree program of TBtools. The green color represents the Mn-SOD group, while the blue color signifies the Fe-SOD group. Lastly, the Zn/Cu-SOD group is represented by the color yellow. (B) Various colored boxes indicate various motifs. Specifically, the pale green color signifies the Sod_Fe_N domain, the golden color signifies the Sod_Fe_C superfamily, the rose color signifies the Sod_Fe_C domain, the azure color signifies the Sod_Fe_N superfamily, the crimson color signifies the Sod_Cu domain, and the violet color signifies HMA. Additionally, the pale green color represents CDS or exons, while the black horizontal line represents introns.

Phylogenetic analysis of BcSODs from B. ceiba and other plants

To analyze the evolutionary relationships between SODs in B. ceiba and other plants, complete BcSOD protein sequences were compared with SOD protein sequences from A. thaliana, Populus trichocarpa, Oryza sativa, and Dimocarpus longan. This comparison was used to construct an unrooted phylogenetic tree (Fig. 5 and Table S3), which indicated that the 46 SODs could be categorized into three separate clusters: Fe-SODs, Mn-SODs, and Cu/Zn-SODs. This division is based on BcSOD domains and is supported by strong bootstrap values.

Figure 5 A maximum phylogenetic tree of 46 SOD proteins from Bombax ceiba, Arabidopsis thaliana, Populus trichocarpa, Oryza sativa, and Dimocarpus longan.

A circular unrooted phylogenetic tree of plant SOD proteins was generated employing the maximum likelihood method through the MEGA 7.0 program. The percentages of replicate trees where the related taxa formed clusters together are indicated by the numbers next to the branches. The unrooted phylogenetic tree was displayed in a circular layout. The names of species consist of the code representing the first letter of the genus, followed by the second letter of the species name. The gene names utilized are At for Arabidopsis thaliana, Pt for Populus trichocarpa, Os for Oryza sativa, DL for Dimocarpus longan, and Bc for Bombax ceiba. Supplemental materials (Table S3) provide accession numbers and sequences pertaining to the anticipated proteins. The Zn/Cu-SODs group is represented by a red line, the Fe-SODs group is represented by a blue line, and the Mn-SODs group is represented by a dark line.

Phylogenetic analysis offered a distinct understanding regarding BcSOD categorization, which was based on association with different metallic coenzymes. The Cu/Zn-SOD group included BcCSD1a, BcCSD1b, BcCSD1c, BcCSD2a, BcCSD2b, BcCSD3, and BcCSDc. BcFSD1, BcFSD2, and BcFSD3 were classified as part of the Fe-SOD group, whereas BcMSD1a, BcMSD1b, and BcMSD2 were placed in the Mn-SOD group (Fig. 5). Importantly, Cu/ZnSODs and Fe/MnSODs were clearly segregated within monocot and dicot clades. Additionally, subcellular forecasts for SODs are likely to help reinforce protein clustering. For example, AtCSD2, BcCSD2a, BcCSD2b, DLCSD2, and PtCSD2 constituted a closely associated group with a significant bootstrap value (100%), and all of these proteins were found in chloroplasts (Table 2).

Protein-protein interaction predictions within the SOD gene family

Our predictions of protein-protein interactions assume independence from gene context methods used by STRING to identify potential interacting proteins. The highest identity homologous AtSOD proteins were designate as STRING proteins based on orthologs in A. thaliana. Figure 6 and Table S5 show that all 13 BcSOD proteins were associated with nine known AtSOD proteins in the interaction network. BcCSD1a, BcCSD1b, and BcCSD1c were linked with AtCSD1; BcCSD2a and BcCSD2b were associated with AtCSD2; and AtCSD3 and AtCSDc were the highest homologous proteins of BcCSD3 and BcCSDc, respectively. For FSDs, BcFSD1, BcFSD2, and BcFSD3 corresponded to AtFSD1, AtFSD2, and AtFSD3, respectively. Additionally, BcMSD1a and BcMSD1b were associated with AtMSD1, and BcMSD2 was associated with AtMSD2. This is consistent with our phylogenetic classification, in which BcSOD proteins corresponding to three types of AtSOD proteins were grouped together in the phylogenetic tree (Fig. 5). The three types of SOD proteins form a robust interaction network, with each BcSOD putatively interacting directly or indirectly with all other BcSOD proteins, and vice versa (Fig. 6). This suggests that protein complexation has a potential regulatory role. Furthermore, the high level of interaction between SODs and catalase indicates a coordinated response of various enzymes within the plant antioxidant enzyme system in response to abiotic and biological stresses. The widespread sub-cellular localization of BcCSD, BcFSD, and BcMSD proteins indicates their reciprocal regulation and co-expression.

Figure 6 Interaction network of BcSODs in B. ceiba.

The higher the interaction coefficient, the darker the red color inside the ellipse, vice versa. BcCSD1a, BcCSD1b, and BcCSD1c were identified as homologous to CSD1 in A. thaliana, BcCSD2a and BcCSD2b to AtCSD2, BcCSD3 to AtCSD3, and BcCSDc to AtCSDc. Additionally, BcFSD1, BcFSD2, and BcFSD3 corresponded to FSD1, FSD2, and FSD3 in A. thaliana. BcMSD1a and BcMSD1b were homologous to MSD1, and BcMSD2 to MSD2 in A. thaliana.

Analysis of cis-regulatory elements in the promoter of BcSODs

The examination of putative cis-acting elements through statistical analysis indicated a notable presence of environmental and hormone-responsive elements dispersed throughout the promoter region of BcSODs (Fig. 7 and Table S6). As depicted in Fig. 7, five phytohormone (salicylic acid (SA), gibberellin (GA), methyl jasmonate (MeJA), auxin, and abscisic acid (ABA)) correlated responsive elements including TGA-element, TGACG-motif, TCA-element, TATC-box, P-box, GARE-motif, CGTCA-motif, ABRE, etc., were identified (Fig. 7 and Table S6). The distribution of these elements varied among genes, emphasizing their pivotal role in phytohormone mediation. Moreover, environmental-responsive elements (drought, anaerobic, low-temperature, light, and defense and stress), including TCT-motif, TC-rich repeats, TCCC-motif, MBS, LTR, LAMP-element, I-box, GT1-motif, G-box, GATA-motif, ARE, AT1-motif, etc., were identified (Fig. 7 and Table S6). Notably, a multitude of light-responsive elements was widely distributed among all promoter region of BcSODs, with BcCSD1b featuring the lowest (11) and BcCSD1c the highest (26) number of light-responsive elements. Additionally, plant hormone-responsive elements (ABA, MeJA, GA, SA, and auxin) exhibited diverse distribution in the promoter region of BcSODs. For instance, BcCSDc featured only two hormone-responsive elements (CGTCA-motif and TGACG-motif) associated with MeJA-responsiveness. BcMSD1a, BcCSD1b and BcCSD1c contained all five types of hormone-responsive elements, while BcFSD1, BcMSD1b, BcMSD2 and BcCSD2b contained four types. Cluster analysis grouped specific BcSODs together, implying potential shared roles in response to similar abiotic stressors or conditions. Specifically, BcCSD1a, BcCSD2a, BcCSD1c, and BcMSD2 formed one cluster, while BcFSD1, BcFSD2, BcFSD3, BcCSDc, and BcCSD3 comprised another. BcMSD1a, BcMSD1b, BcCSD1b, and BcCSD2b fell into the same subgroup. These clusters suggest potential functional similarities among the grouped BcSODs. Overall, the results suggest that the expression levels of BcSODs may diverge under phytohormone and abiotic stress conditions.

Figure 7 Analysis of cis-regulatory elements in the 3 kb region upstream of the ATG (promoter) of BcSODs.

The cluster heatmap was drawn using the plug-in heatmap of TBtools software. The number inside the rectangular represents the number of cis-regulatory elements contained.

Responses of BcSOD expression to AMF inoculation and drought stress

We measured the impact of drought stress and AMF on the mRNA levels of BcSOD. All seven Cu/ZnSODs in the root were significantly upregulated by AMF colonization under drought conditions, except for BcCSD3 (Fig. 8A). Furthermore, AMF colonization demonstrated a positive impact on the expression of BcMSD1a and BcMSD1b, while it led to a decrease in BcFSD1 transcript levels in the root under drought stress conditions (Figs. 8A and 8B). In well-watered conditions, AMF colonization exhibited a dual regulatory effect in root tissues: upregulation of BcCSDs and downregulation of BcFSD1 expression (Figs. 8C and 8D). In shoots tissues under drought stress, all seven BcCSDs were upregulated by AMF inoculation, but there was no significant effect on the BcFSDs and BcMSDs group (Fig. 8C). Under well-watered conditions, the impact of AMF colonization was relatively modest, except for the notable upregulation of BcCSD1a, BcCSD2a, and BcFSD2 in shoots tissues (Fig. 8D). In summary, AMF colonization emerges as a key modulator, particularly enhancing the expression of Cu/ZnSODs during drought stress in both root and shoot tissues.

Figure 8 The expression pattern of 13 BcSOD genes in the roots and shoots of B. ceiba seedlings subjected to AMF and drought stress treatments.

(A) Relative expression of SODs in root under drought stress.(B) Relative expression of SODs in shoot under drought stress. (C) Relative expression of SODs in root under well watered treatment. (D) Relative expression of SODs in shoot under well watered treatment. Note: The data represents the means of four biological replicates ± standard deviation (n = 4). Student’s T test was conducted to evaluate the significant difference between AMF and NMF treatments. *: 0.01 < P < 0.05, **: 0.001 < P < 0.01, ***: P < 0.001. NS, no significant. AMF indicate seedlings colonized with R. irregularis treatment, NMF indicate seedlings mock colonized with R. irregularis.

Discussion

Drought stress poses a significant abiotic challenge to plants globally (Piao et al., 2010). The symbiotic association of plants with AMF has been acknowledged for its vital function in alleviating plant drought stress through the modulation of diverse biochemical and physiological pathways (Bahadur et al., 2019; Mitra et al., 2021; Li et al., 2022). During our investigation, mycorrhizal B. ceiba demonstrated better development compared to non-mycorrhizal seedlings in both DS and WW (Fig. 2). This finding underscores the positive impact of AMF on B. ceiba’s drought tolerance.

Drought stress triggers photosynthetic dysfunction in plants, disrupting processes such as gas exchange, light harvesting, and chloroplast photochemistry. Such disruption increases ROS formation and consequent oxidative damage (Hasanuzzaman et al., 2020). In our prior investigation, we noted a marked increase in the speed of O2− production during drought in B. ceiba (Li et al., 2022). The O2−/H2O2 system, which catalyzes the transformation of ROS into harmless compounds, is a vital route for scavenging ROS (Sies, Berndt & Jones, 2017). SOD has a key role in this process by facilitating the conversion of O2− to H2O2 and O2 (Hodgson & Fridovich, 1973; Sies, Berndt & Jones, 2017; Choudhary et al., 2018). This function was improved by AMF inoculation when B. ceiba seedlings were exposed to drought stress (Fig. 3). These results suggest that AMF inoculation enhances plants’ ability to scavenge O2− during periods of drought stress.

Consistent with our measurements of SOD activities, root and shoot expression levels of all Zn/Cu-SODs were higher in AMF compared to NMF in the DS treatment group, except for BcCSD3 in roots. Similarly, He, Sheng & Tang (2017) reported elevated transcript levels of Cu/Zn-SOD and lower concentrations of O2−, H2O2, and MDA in both leaves and roots of mycorrhizal black locust under drought conditions. Furthermore, root BcMSD1a and BcMSD1b transcript levels were higher in AMF under DS (Fig. 7). Our findings are also aligned with the results of other work by He et al. (2020), which found that the relative expression of PtMn-SOD was elevated in trifoliate orange leaves under drought stress in plants colonized by AMF. The same study reported down-reguation of Fe-SOD gene expression in leaves, which is also consistent with our finding that the relative expression of BcFSD1 was consistently lower in roots in the AMF treatment group, regardless of soil moisture regime (Fig. 7). This implies that higher SOD activity in mycorrhizal B. ceiba seedlings may result from a combination of specific isozymes and the expression of genes encoding SODs (Ruiz-Lozano et al., 2001). Mycorrhizal modulation of SOD gene expression appears to depend on SOD isozyme gene types. In contrast to our observations of SOD activities in roots under WW, the relative expression levels of all Zn/Cu-SODs were higher in AMF-colonized roots compared to NMF, except for BcCSD1b. This may be the result of the symbiotic relationship between plant roots and AMF, which increases the expression of SOD genes. He et al. (2020) found thatCu/ZnSOD gene expression was significantly upregulated in response to fungal colonization, independent of soil moisture regime. The expression pattern we observed is likely associated with host plant regulation of intracellular AMF colonization (Salzer, Corbière & Boller, 1999). Functional Cu/Zn-SOD genes could be part of a defense mechanism that neutralizes ROS and counters localized host defense responses present in arbuscule-containing cells within the germinated spores of the AMF Gigaspora margarita (Lanfranco, Novero & Bonfante, 2005). These findings suggest that AMF colonization augments antioxidant defense mechanisms, particularly by modulating SOD activities and gene expression, thereby helping alleviate oxidative stress in B. ceiba seedlings experiencing water limitation. Nevertheless, field experiments and additional laboratory experiments assessing the effects of a broader range of AMF species are needed to further clarify the impact of AMF on drought tolerance in B. ceiba.

In this study, we identified 13 BcSOD genes, comprising seven Cu/Zn-SODs, three FeSODs, and three Mn-SODs (Table 2), and clustered them into three major groups based on their metal-binding domains (Fig. 4). Consistent with evolutionary trends observed in most angiosperms, the number of BcSOD family members remained highly conserved. Such conservation is evident in various plant species, such as Akebia trifoliata and maize, which both have 13 SOD genes (Liu et al., 2021; Yang et al., 2023); Medicago truncatula and barley, with seven (Song et al., 2018; Zhang et al., 2021); tomato with nine (Feng et al., 2016); grapevine with ten (Hu et al., 2019); cotton with 18 (Wang et al., 2017); and wheat with 26 (Jiang et al., 2019). Gene structures were highly conserved across the three BcSOD subfamilies (BcFSDs, BcMSDs, and BcCSDs). The protein sequences in these subfamilies displayed different motifs, reflecting a bias observed in their compositions (Fig. 4B). The patterns of introns and exons, which are indicators of gene evolution, have changed only slightly in BcSOD genes, particularly within the same subfamily. This is consistent with patterns observed in the SOD genes of other plant species, such as Brassica napus and Akebia trifoliata (Su et al., 2021; Yang et al., 2023). The shared exon–intron associations and conserved motifs among SOD genes within each cluster further suggest that they mediate responses to various abiotic stressors in similar ways, a pattern previously reported in tomato, cotton, tea, and wheat (Feng et al., 2016; Wang et al., 2017; Jiang et al., 2019; Zhou et al., 2019).

The strategic positioning of SOD proteins within cellular organelles is central to plant responses to oxidative stress induced by abiotic factors, facilitating cellular signal conversion (Del Río et al., 2018). Localization predictions revealed that three members (BcCSD1a, BcCSD1b, and BcCSD1c) were located in the cytoplasm; five members (BcFSD1, BcFSD2, BcFSD3, BcCSD2a, and BcCSD2b) in chloroplasts; all three MnSODs in mitochondria; BcCSDc in the peroxisome; and BcCSD3 in the cytoskeleton. These findings are consistent with previous work reporting the localization of Fe-SODs in chloroplasts and Mn-SODs in mitochondria and peroxisomes (Abreu & Cabelli, 2010; Perry et al., 2010). Meanwhile, Cu/Zn-SODs were predominant and primarily found in chloroplasts, cytoplasm, and peroxisomes (Song et al., 2018). In addition, heatmap clustering analysis indicated differential expression of SOD genes in different tissues. Specifically, transcription levels of BcCSD1a, BcCSD1b, BcCSD1c, and BcMSD1a were higher in shoots than in roots. Conversely, transcription levels of the remaining genes were higher in roots than in shoots, particularly when colonized by AMF. Our findings provide strong evidence that AMF colonization modifies tissue expression patterns of SOD genes.

To understand the roles of BcSOD genes in drought stress response, we predicted cis-elements in the genes’ promoter regions. Our results revealed three types of cis-elements: environmental-responsive, hormone-responsive, and other (Fig. 6 and Table S5). Most cis-elements were associated with SA, MeJA, ABA, GA, light, anaerobic, drought, and low temperature induction. The presence of abundant environmental and plant hormone-responsive elements in BcSOD promoters suggests potential association with mechanisms regulating responses to both abiotic and biotic stress. Previous work has established the importance of cis-elements to plant stress responses (Maruyama-Nakashita et al., 2005; Osakabe et al., 2014). Together with these findings and studies in different plants, our work highlights the important role of SOD genes in mediating plant responses to diverse stressors (Jiang et al., 2019; Su et al., 2021; Yang et al., 2023). Furthermore, there is considerable diversity in both the quantity and category, and certain components that react to metabolic processes and genetic manifestation pertain exclusively to certain SOD genes. These findings enhance our understanding of how BcSOD genes respond to various environmental conditions.

Conclusions

Here, we isolated 13 BcSOD genes from B. ceiba, with subsequent phylogenetic analysis demonstrating that they are conserved. This conservation was further corroborated by motif and conserved domain analyses. We postulate that AMF symbiosis alleviates the negative effects of drought stress on B. ceiba seedlings by changing the expression patterns of SOD genes. The relative expression levels of SOD genes were significantly higher in AMF-colonized plants under drought stress, driving higher SOD activity and improving ROS delimitation (i.e., scavenging). The AMF R. irregularis improves the drought resistance of B. ceiba by regulating SOD activities and SOD gene expression. To our knowledge, this is the first report on the effect of AMF on SOD expression under drought stress at the whole genome level in B. ceiba. Our study provides molecular evidence for the mechanisms underlying the improved drought resistance resulting from AMF colonization and yields insights into SOD gene expression in B. ceiba, thereby supporting work to improve mycorrhizal seedling cultivation under stress.

Supplemental Information

Supplemental Information 1 MIQE checklist.

Supplemental Information 2 Raw data.

Supplemental Information 3 Supplementary figures and tables.

Figure S1: (a) Multi-alignment of the Fe/Mn-SOD protein sequences; (b) Multi-alignment of the Zn/Cu-SOD protein sequences. Conserved domains in alignment are highlighted with colored lines, and their names are given above the conserved regions; Figure S2: Heatmap and cluster analysis of 14 BcSOD genes relative expression in seedlings that received the AMF and drought stress treatments.

Additional Information and Declarations

Competing Interests

Author Contributions

Data Availability

The authors declare that they have no competing interests.

Changxin Luo performed the experiments, prepared figures and/or tables, and approved the final draft.

Zhumei Li performed the experiments, authored or reviewed drafts of the article, and approved the final draft.

Yumei Shi analyzed the data, authored or reviewed drafts of the article, and approved the final draft.

Yong Gao analyzed the data, authored or reviewed drafts of the article, and approved the final draft.

Yanguo Xu performed the experiments, analyzed the data, prepared figures and/or tables, and approved the final draft.

Yanan Zhang conceived and designed the experiments, authored or reviewed drafts of the article, and approved the final draft.

Honglong Chu conceived and designed the experiments, authored or reviewed drafts of the article, and approved the final draft.

The following information was supplied regarding data availability:

The raw measurements are available in the Supplemental Files.

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
