# Peer review of "Arbuscular mycorrhizal fungi enhance drought resistance in Bombax ceiba by regulating SOD family genes"

_PeerJ, doi:10.7717/peerj.17849_

## Round 0.1 · original submission · Major Revisions

· Academic Editor

Major Revisions

All the reviewers agree the paper has significant applied value. The first and third Reviewers have minor comments about the presentation of the paper, please attend to these. Reviewer 2 has more substantive comments about how to communicate the experiment better, please pay particular attention to these.

**Language Note:** The review process has identified that the English language must be improved. PeerJ can provide language editing services - please contact us at [email protected] for pricing (be sure to provide your manuscript number and title). Alternatively, you should make your own arrangements to improve the language quality and provide details in your response letter. – PeerJ Staff

·

Basic reporting

The manuscript entitled as “Arbuscular Mycorrhizal Fungi Enhanced Drought Resistance of Bombax ceiba by Regulating the SOD Family Genes” highlights the importance of arbuscular mycorrhizal (AM) fungi for contributing to the plants' ability to tolerate drought. Authors have performed pot trial to examine if the arbuscular mycorrhizal fungus (Rhizophagus irregularis) improved the drought resistance of Bombax ceiba. The SOD gene family underwent a comprehensive analysis in this research, and the genome wide expression patterns of SODs and the SOD activity were evaluated in AMF and NMF plants under drought.
The research is conducted systematically and followed the standard scientific methods. In the genome of B. ceiba, a total of 13 superoxide dismutases (SODs) were found, including three FeSODs (BcFSDs), three MnSODs (BcMSDs), and seven Cu/ZnSODs (BcCSDs). The present study highlighted many innovative gene regulatory mechanisms of plants in response to drought i.e., hormones and stress-responsive cis-regulatory elements were found in the promoters of every BcSODs.
Further results described such as expression proûling using qRT-PCR confirmed that AMF enhanced the relative expression levels of Cu/Zn-SODs in both roots and shoots during drought stress, except for BcCSD3 in the root. Furthermore, AMF additionally increased the relative expression of BcMSD1a and BcMSD1b in the root, leading to an augmentation in SOD activity and improvement in ROS scavenging abilities during periods of drought stress.
Overall the present study is quite important in its scientific direction and can be published in PeerJ journal but I have major concerns that authors need to consider and revise their publication accordingly.

Experimental design

Materials and Methods
Experimental design
Line 123-128 “The study utilized a fully randomized block design, incorporating two variables: (1) AMF treatments, involving the introduction of Rhizophagus irregularis (Blaszk, Wubet, Renker & Buscot) Walker & Schüßler (BGC BJ09) (+AMF) and the absence of (-AMF) R. irregularis, and (2) soil water regimes, encompassing well-watered (WW, maintaining 60% of the substrate's maximum water holding capacity) and drought stress (DS, maintaining 30% of the substrate's maximum water holding capacity) conditions”.
These sentences are not clear. Please rewrite again. Please make them shorter that description should be clearly understandable.
Line 182 “magnified cross-sections was utilized (McGonigle et al. 1990).”
Was should be replaced by were
Line 200-207 “All candidate sequences meeting the criteria were further validated through the InterPro (https://www.ebi.ac.uk/interpro/) database, the Hammer SMART (http://smart.embl-heidelberg.de/#) database, and NCBI Conserved Domain Search (https://www.ncbi.nlm.nih.gov/Structure/cdd/wrpsb.cgi) for conserved domain verification, specifically checking for the presence of SOD domains such as SOD_Fe_C-terminal (PF02777), Iron/manganese superoxide dismutases, SOD_Fe_N-alpha-hairpin (PF00081), and Copper/zinc superoxide dismutase (SODC) (PF00080.23) were considered”.
There are many grammatical errors are in these sentences. Please rewrite them again.
Line 298 “The instability index revealed that, except for BcFSD1 and BcFSD1”, This sentence is confusing. Please make it clear.

Validity of the findings

Overall the present study is quite important in its scientific direction and can be published in PeerJ journal but I have major concerns that authors need to consider and revise their publication accordingly.
Abstract
Line 21-22 “mycorrhizal fungus (Rhizophagus irregularis) improved the drought resistance of Bombax ceiba,” Here improved should be replace by improve.
Introduction
Line 41 “Global Global climate change is causing more frequent and severe drought stress,”
Please remove one Global

Additional comments

Discussion
Line 361-404 Please divide this into 2 to 3 small paragraphs.
Figure 2, Figure 3, and Figure 8, Why not the bar graphs should have the same color. Please make similar color for the bar graphs in mentioned Figures.
Please add some more references such as

“The multifaceted role of sodium nitroprusside in plants: crosstalk with phytohormones under normal and stressful conditions” by Ullah et al., 2024 in Plant Growth Regulation https://doi.org/10.1007/s10725-024-01128-y.
“Response of source-sink relationship to progressive water deficit in the domestication of dryland wheat” by Gui et al., 2024 in Plant Physiology and Biochemistry https://doi.org/10.1016/j.plaphy.2024.108380
“Physiological and biochemical responses of two spring wheat genotypes to non-ydraulic root-to-shoot signalling of partial and full root-zone drought stress” by Batool et al., 2019 in Plant Physiology and Biochemistry https://doi.org/10.1016/j.plaphy.2019.03.001

Reviewer 2 ·

Basic reporting

This manuskript provides an intriguing study on the positive effects of arbuscular mycorrhizal fungi (AMF) in boosting drought resistance in Bombax ceiba by altering the expression of superoxide dismutase (SOD) genes. The authors did an extensive work, using genomic and physiological analysis to explore the complex interactions between AMF and plant's response to stress. They've identified 13 SOD genes and thoroughly analyzed their expression under drought situations, offering deep insights into the mechanisms behind the tolerance induced by AMF to stress. However, this research could be even stronger with a more detaild statistical analysis, like explaining why certain sample sizes were choosen. This is especially important for such complex biological interactions as those between plants and fungi under drought. Also, clarifying the normalization methods used in the qRT-PCR data for gene expression would be beneficial, particularly how the reference gene(s) for normalization were picked and their consistency checked across different conditions. This is crucial for validating gene expression data. Moreover, making the figures clearer and providing more info on experimental methods would make the paper easier to read and replicate. Adding these improvements would make the study even more solid and accessible.

Experimental design

1. The manuskript could really use some explanation or justification for the sample size used. Knowing how the sample size was decided upon, including any power analysis done to make sure the study was powerful enough to detect a meaningful effect, would make readers more confident in the results. This detail is very important, especially when you're looking at complex biological interactions like those between plants and fungi under drought conditions.
2. For the qRT-PCR data on gene expression, a more detailed explanation of the normalization methods would help. It would be nice to know how the reference gene(s) for normalization were selected and their stability verified across the various experimental setups. This info is vital for the reliability of gene expression data, particularly when deducing the significance of the observed expression patterns.
3. Although the figures are informative, making them clearer to show gene expression data could greatly help. Like, adding error bars to show variability among replicates and using color codes or different patterns to differentiate between treatments (AMF vs. NMF) in various drought scenarios would help in quickly understanding the presented data.
4. Choosing the right keywords can majorly affect how easy it is to find the research. Consider adding more specific terms related to the studied mechanisms, like "reactive oxygen species (ROS) scavenging" or "drought stress signaling," to draw in readers interested in the molecular and physiological details of how AMF helps with drought resistance.
5. The manuscript would be better off with a clearer statement of the research hypothesis early on. Specifically, how does AMF colonization affect SOD gene expression in Bombax ceiba under drought conditions? Clarifying this early in the manuscript would help readers grasp the flow of the research and its importance.

Validity of the findings

1. More details on how drought stress was quantitatively set up and monitored throughout the study would be useful. Information like how 30% of the substrate's maximum water holding capacity was chosen as the drought condition and how this level was consistently maintained across the experiment is key for reproducibility.
2. Expanding the methods section to include more details about the AMF inoculation process would improve the manuscript. Describing how the purity and viability of the Rhizophagus irregularis inoculum were checked before its use to make sure the observed effects are due to AMF colonization would be useful.

Reviewer 3 ·

Basic reporting

This manuscript describes the effects of Arbuscular Mycorrhizal Fungi on Drought Resistance of Bombax ceiba by Regulating the SOD Family Genes. Overall, the results support that this treatment was effective, and the comprehensive analysis of the SOD gene family helps to explain the mechanistic components of these beneficial effects. As such, the research seems appropriate for this journal; however, the following issues will need to be addressed before it is considered for publication.

These are my suggestions for the authors:

1.Check the spelling and grammar of the entire manuscript.
2.Line 19-23: Please improve this paragraph by rearranging the structure to make the main objective of the study clear.
3.The introduction section, in its current form, is too long and difficult to follow. Therefore, I suggest shortening this section. Additionally, some other comments in this section are listed below:
4.Line 86-87: Please rewrite, as it is not clear.
5.Line 89: Please remove "Symbiotic."
6.Line 92-94, 97-98: This part is unclear. Please rewrite the text.
7.Line 112-119: Clearly state why the authors conducted this study and how it differs from previous research. Provide the logical background/significance to support your investigation.
8.Line 167: Specify whether the measurements were taken from all seedlings or a subset of them.
9.Line 167: Please add "the" before "height."
10.Line 169-170: Replace "In order to examine AMF colonization, the root washed by tap water and a little of them were conserved in FAA fixative" with "To examine AMF colonization, the roots were washed with tap water, and a little of them were conserved in FAA fixative."
11.Line 179: Please remove "the" before "trypan."
12.Line 271: Please remove "the" before "well-watered."
13.Line 321: Please add "a" before "distinct."
14.Line 354: Replace with "The symbiotic association of plants with arbuscular mycorrhizal fungi (AMF) has been acknowledged for its vital function in alleviating plant drought stress through the modulation of diverse biochemical and physiological pathways."
15.Line 356-359: Please rephrase.
16.Line 379: Please add "the" before "study."
17.The legends for Figure 1 need to be elaborated upon; please revise.
18.The legends for Table 1 need to be elaborated upon; please revise

Experimental design

no comments

Validity of the findings

no comments

---

## Round 0.2 · accepted · Accept

· Academic Editor

Accept

The Reviewers all confirm that you have made a strong effort in addressing their concerns, and the paper is now publishable.


·

Basic reporting

No Comment

Experimental design

No Comment

Validity of the findings

No Comment

Additional comments

All the comments have been carefully revised and Authors have improved the writing as well as presentation of Data in Figures and Tables. I am satisfied with the revisions and the manuscript can be accepted for publication in present form.

Reviewer 2 ·

Basic reporting

None

Experimental design

None

Validity of the findings

None

Reviewer 3 ·

Basic reporting

I have read the revised version with interest. The authors have exerted effort in addressing my comments. The paper is now publishable

Experimental design

no comment

Validity of the findings

no comment

Additional comments

no comment